# SAM2Flow: Interactive Optical Flow Estimation with Dual Memory for *in vivo* Microcirculation Analysis

**Luojie Huang[1]    Ryan Zhang[1]    Marisa M. Morakis[1]    Michaela Taylor-Williams[1]**
**Gregory N. McKay[1]    Nicholas J. Durr[1,†]**
[1]Department of Biomedical Engineering, Johns Hopkins University, Baltimore, USA
`ndurr@jhu.edu`

## Abstract

Analysis of noninvasive microvascular blood flow can improve the diagnosis, prognosis, and management of many medical conditions, including cardiovascular, peripheral vascular, and sickle cell disease. This paper introduces SAM2Flow, an interactive optical flow estimation model to analyze long Oblique Back-illumination Microscopy (OBM) videos of *in vivo* microvascular flow. Inspired by the Segment Anything Model (SAM2), SAM2Flow enables users to specify regions of interest through user prompts for focused flow estimation. SAM2Flow also incorporates a dual memory attention mechanism, comprising both motion and context memory, to achieve efficient and stable flow estimations over extended video sequences. According to our experiments, SAM2Flow achieves SOTA accuracy in foreground optical flow estimation on both microvascular flow and public datasets, with a fast inference speed of over 20 fps on $512 \times 512$ inputs. Based on the temporally robust flow estimation, SAM2Flow demonstrated superior performance in downstream physiological applications compared to existing models. The code is available at: `https://github.com/DurrLab/SAM2Flow`.

## 1   Introduction

Microvascular blood flow parameters, such as flow velocity, provide critical insight to understand and manage diseases that affect blood rheology and vascular mechanics. Noninvasive microvascular imaging allows visualization of vessel and blood cell dynamics at the single-cell level *in vivo*, providing a window into the early progression of diseases and the real-time rheological status of the patient. Some applications include measuring microvascular elasticity to diagnose coronary microvascular dysfunction [1–3], monitoring tumor angiogenesis to assess cancer progression and treatment response [4], and evaluating blood cell function in sickle cell disease [5, 6].

Established techniques for measuring vascular flow include Laser Doppler Flowmetry (LDF) [7] and Laser Speckle Contrast Imaging (LSCI) [8]. But these techniques acquire low-resolution data and are most sensitive to arterioles and venules, which are much larger and deeper beneath the skin compared to capillaries. Orthogonal Polarization Spectral (OPS) imaging [9] and Side-stream Dark Field (SDF) imaging [10] have also been explored for capillary-resolution microvascular measurement of the human oral cavity and finger nailfold. Recently, Oblique Back-illumination Microscopy (OBM) [11] has demonstrated the potential for non-invasive microvascular measurements with high-speed imaging and subcellular resolution. Green-light OBM can simultaneously record phase and absorption contrast from individual blood cells flowing through superficial capillaries [12].

For all microvascular imaging techniques, quantitative, accurate, and efficient video analysis is critical to enable clinical impact. Current analysis approaches often use semiquantitative metrics

---

[†]Corresponding author.

39th Conference on Neural Information Processing Systems (NeurIPS 2025).

as the Microvascular Flow Index (MFI) [13], or automated spatiotemporal diagram analysis [14], which are labor-intensive, requiring extensive pre-processing, including background correction, video stabilization, vessel segmentation, and manual refinement. While a cell tracking model [15] has been explored for flow characterization, the computational cost increases drastically for longer videos or large vessels packed with more cells, such as arterioles and venules. To achieve meaningful clinical impact with microvascular flow technologies, efficient, fully automatic flow estimation is critical.

Optical flow (OF) models predict the motion of objects in a video by estimating 2D vector fields that represent pixel-wise displacements between consecutive frames. Despite the impressive performance of deep learning-based OF models in various applications, their application to *in vivo* microvascular flow estimation remains unexplored. Existing models, such as RAFT [16] and GMA [17], are mostly constrained to the analysis of two frames or short segments. More recent models incorporate mechanisms for longer sequences [18, 19], such as memory [20]. However, challenges persist in achieving robust flow estimation over long videos. Moreover, conventional OF models estimate optical flow maps across all pixels, which is unnecessary when the background movements are not relevant. Lastly, the limited availability of high-quality datasets, especially non-synthetic videos, also hinders the adaptation of OF neural networks to various domains.

The Segment Anything Model (SAM) [21] and its successor SAM2 [22] were introduced as foundation models for instance segmentation in images and videos. Pre-trained with an unprecedentedly large dataset, SAM2 demonstrated superior scene and object understanding, as well as object memory across frames. Although these models are trained on natural scene datasets, impressive generalizable performance has been achieved by initializing with these pre-trained weights and fine-tuning with task- and modality-specific datasets, particularly in the medical field [23–25].

To bridge the gap between the latest deep learning research for video analysis and non-invasive blood flow estimation, we introduce SAM2Flow, an optical flow estimation model for vessel-emphasized blood flow analysis of long *in vivo* OBM videos. Inspired by SAM2, SAM2Flow accepts user prompts to select the regions of interest (ROI), such as target capillary segments or specific branches of a complex vascular structure, for fine-grained flow estimation. To ensure robust flow estimation across long videos, SAM2Flow also incorporates a dual memory mechanism, comprising both motion memory and context memory. Fig. 1 showcases the improved clinical workflow by incorporating SAM2Flow as the efficient end-to-end microvascular flow estimation model.

SAM2Flow represents five major contributions to the field of optical flow estimation and microvascular flow analysis: 1) **Introduction of SAM2Flow**: The first optical flow neural network specifically designed for blood flow estimation. 2) **Interactive ROI optical flow estimation**: Enables user-guided selection of regions of interest for efficient flow analysis. 3) **Dual memory mechanism**: Incorporates motion and context memories to ensure efficient and stable flow estimation in long video sequences. 4) *In vivo* **blood flow dataset for optical flow estimation**: Establishes a large human capillary flow video dataset with paired optical flow maps to facilitate future research. 5) **General-purpose foreground optical flow model**: The experiment on the public benchmark, Spring, demonstrates the promising performance of SAM2Flow on joint motion foreground detection and ROI-centric optical flow estimation, beyond the microscopic domain.

## 2 Related Works

### 2.1 *In Vivo* Blood Flow Estimation

Superficial microvascular measurement is critical to study skin perfusion [26], wound healing [27], peripheral vascular diseases [28], and neurological blood flow changes [29]. LDF [7] is a widely adopted non-invasive technique to estimate blood flow by measuring the Doppler shift of a low-power laser beam source caused by moving RBCs. The result is typically reported in units of perfusion, a combination of RBC concentration and velocity. Another popular imaging-based technique, LSCI [8], maps perfusion by the blur of coherent laser speckle grains from light-scatter of flowing red blood cells. Both methods qualitatively measure flow and typically display in units of relative blood flow instead of absolute flow velocity. In addition, the limited spatial or temporal resolutions of these techniques hinder their ability to localize blood flow changes in specific microvessels. Lastly, both techniques are very sensitive to patient or probe motions. Alternative techniques for flow perception use special sensors, such as event cameras [30] and spike cameras [31], but the limited quality of the reconstructed frames hampers the visualization of clinically relevant anatomical details.

More recent flow estimation techniques using OPS [9] and SDF [10] image superficial capillaries at various sites, including oral cavity, nailfold, and retina. With these techniques, microvascular flow can be differentiated from the background tissue due to the strong green-light absorption of hemoglobin in red blood cells. Previous flow estimation work relied on the movement of absorption gaps from transparent white blood cells (WBCs) or plasma [32, 33]. A semiquantitative metric, MFI [13], characterizes microcirculation status in OPS or SDF capillary videos. Estimations are achieved by subjectively classifying each flow as absent (0), intermittent (1), sluggish (2), or smooth (3). The video-based MFI is calculated as the average score over all labeled FoV quadrants or vessels. Spatiotemporal(ST) diagrams [14] were applied to absorption-based blood flow videos as an automatic evaluation method. The flow can be quantified by the tilted angle of lines in the ST diagram. Bourquard *et al.* estimated flow speed and detected WBC from nailfold flows using this method [34]. However, continuous blood flow estimation is hard due to the scarcity of absorption gaps in blood. WBCs typically only account for 0.1% to 0.2% of total blood cells [35]. Moreover, WBC motion is unreliable in representing the net blood flow, due to their unique behaviors, such as rolling and adhesion along the endothelium for leukocyte recruitment during inflammation [36].

OBM has recently been applied to *in vivo* microcirculation measurement [12]. In addition to absorption contrast, OBM introduces phase contrast, resulting in enhanced visualization of the boundary membranes of both red and white blood cells. Deep learning-based models have been explored to achieve cytometry and flow estimation on OBM videos. CycleTrack [15], a multi-object tracker (MOT), showed promising performance detecting and tracking individual cells throughout the video. One major limitation is that the computational cost increases drastically as the video gets longer or more cells are presented in the FoV. Therefore, an efficient end-to-end algorithm is needed for imaging-based *in-vivo* blood flow estimation.

## 2.2 Optical Flow Estimation

**Two-frame optical flow** is traditionally done through the use of optimizing energy functions to maximize the similarity between two images [37–41]. Most current OF research uses deep learning techniques to predict pixel level movement from one image to the next. FlowNet, one of the pioneering works in applying convolutional neural networks (CNNs) to OF, introduced an end-to-end trainable framework that demonstrated the viability of learning-based motion prediction [42]. Building upon the use of CNNs, RAFT utilizes a Convolutional Gated Recurrent Unit (ConvGRU), which allows iterative refinement of the flow output from a multi-scale 4D cross-correlation volume [16]. Since RAFT was published, works such as GMA, Flowformer++, MatchFlow, and SEA-RAFT have all built upon and improved either the training or architecture of the RAFT base [17, 43–46]. Our method, SAM2Flow, harnesses the idea of iterative refinement but does so under explicit segmentation-aware cues as well as a dual memory mechanism to ensure flow consistency even across a wide range of time steps.

**Video-based optical flow** is a method in which multiple frames are used as the input to predict the optical flow at a time point. PWC-Fusion fuses the past flow estimates to the current frame by warping them via a small network [47]. This backward-flow fusion of past frames provides additional longer-term motion cues but yields only modest accuracy gains ( 0.65% improvement over two-frame PWC-Net). In contrast, TransFlow and VideoFlow explicitly leverage a wider temporal window by processing a five-frame window centered on the flow prediction [18, 48]. TransFlow employs a purely transformer-based architecture with a spatio-temporal encoder that attends across patches of all input frames, capturing long-range correlations, and a decoder that uses the combined feature maps from multiple frames to predict the flow [48]. VideoFlow, on the other hand, uses a TRi-frame Optical Flow (TROF) module that jointly estimates bi-directional flows from a center frame to its previous and next frames. A Motion Propagation (MOP) module then links these tri-frame units, propagating motion features so that the effective temporal receptive field grows to cover long sequences. The benefits of higher performance come at a computational cost, jointly modeling multiple frame incurs a large memory footprint and computational overhead. Both TransFlow and VideoFlow require access to future frames and run significantly slower than comparable two-frame models. StreamFlow [19] improves computation efficiency by eliminating redundant processing through the non-overlapping Streamlined In-batch Multi-frame (SIM) pipeline. However, the fixed temporal window remains limited to only a few frames, restricting its ability to maintain stable long-term estimations.

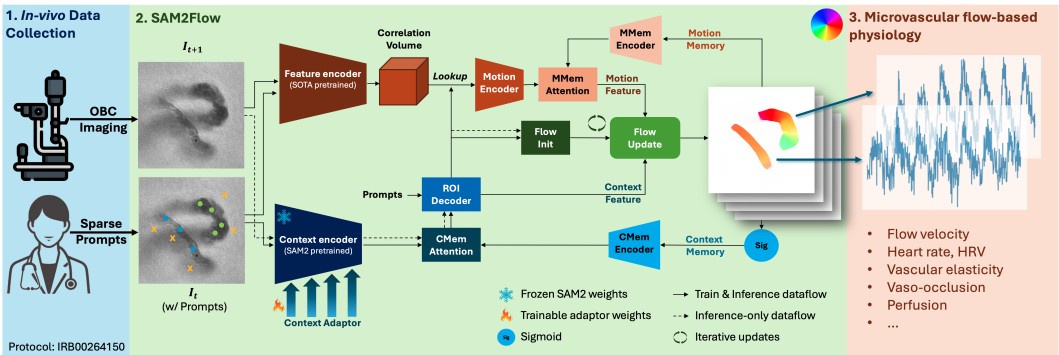

Figure 1: **Clinical workflow of non-invasive microvascular flow evaluation integrating SAM2Flow**, the end-to-end optical flow estimation model. Prompted by sparse point labels, SAM2Flow aims for robust flow estimation in detected ROIs, over long *in-vivo* microvascular flow sequences with the dual-memory mechanism, including context and motion memories. The flow estimation output is critical for various physiological measurements.

To address these drawbacks, MemFlow introduces an efficient memory-based design that processes videos in an online fashion [20]. Instead of stacking multiple frames into the network at once, MemFlow maintains a learned memory bank of frame-wise motion features and context embeddings, which is queried (via attention) to provide relevant past motion cues for the current frame. This design effectively captures long-range temporal information without needing to explicitly feed in a long frame sequence, dramatically reducing computation per flow frame. While all of these methods highlight the benefits of multi-frame input optical flow for maintaining consistency across time, ablation studies in MemFlow found that models do not meaningfully utilize extended histories, with less historical frame data leading to higher performance[20]. In the blood flow estimation task, objects move much faster across frames compared to traditional optical flow tasks, even when captured at 200 FPS. This results in fast-moving, optically ambiguous data that requires memory of previous flow patterns to maintain consistency. SAM2Flow seeks to solve this problem through a dual memory mechanism that ensures stable optical flow over long sequences.

## 3 SAM2Flow

### 3.1 Definition and Overview

**Problem Setup.** Forward optical flow is a per-pixel 2D displacement vector field: $f_{t \to t+1} = (f^u, f^v) \in \mathbb{R}^{H \times W \times 2}$, mapping the location $(u, v)$ from the current frame $I_t \in \mathbb{R}^{H \times W \times 3}$ to the location $(u + f^u(u,v), v + f^v(u,v))$ in the next frame $I_{t+1}$.

Mainstream optical flow networks comprise three major modules: a feature encoder, $\mathcal{E}^f(I_t), \mathcal{E}^f(I_{t+1}) \in \mathbb{R}^{h \times w \times D_f}$, that extracts a pair of low-level textural features to match across frames; a context encoder, $\mathcal{E}^c(I_t) \in \mathbb{R}^{h \times w \times D_c}$, that encodes high-level current context to ensure a meaningful and smooth optical flow field; and a flow update module, $\Delta f = \mathcal{U}(\mathcal{E}^c(I_t), m_t)$, which fuses context understanding with encoded motion features and iteratively refines optical flow estimation with an RNN-based model. The image feature pair is used to construct a series of 4D correlation volumes, $\mathcal{C}_k = \mathcal{E}^f(I_t) \odot \text{AvgPool}(\mathcal{E}^f(I_{t+1}))^\top \in \mathbb{R}^{h \times w \times \frac{h}{2^k} \times \frac{w}{2^k}}$, where $\odot$ represents the correlation operator which computes similarities (dot products) between each pixel in the image feature pair. $\mathcal{C}$ is then used to calculate motion features, $m_t = \mathcal{E}^m(\text{LookUp}(\mathcal{C})) \in \mathbb{R}^{h \times w \times D_m}$. The LookUp operator is some function that returns a motion feature vector for each pixel in $I_t$.

**Overview.** The overview of SAM2Flow is presented in Fig. 1. We used the SEA-RAFT feature encoder, a ResNet-based encoder pre-trained on 6 different optical flow datasets, as the feature encoder backbone $\mathcal{E}^f(\cdot)$. As the context encoder backbone $\mathcal{E}^c(\cdot)$, we integrate the SAM2 ViT image encoder for its strong semantic feature encoding and generalizability from its large-scale pretraining. To adapt the SAM2 encoder to the OBM domain with limited GPU resources, we apply the SAM2 adapter [49], which injects fine-tuning weights into the frozen encoding trunk layers. We choose ConvNeXt-RNN from SEA-RAFT as our flow updates backbone for its superior efficiency in flow

regression. We will elaborate on our proposed prompt-guided flow estimation in section 3.2 and dual memory module for optical flow estimation in section 3.3. We apply two efficient flow initialization methods, **ROI Registration** and **Warm Start**, to further facilitate flow estimation. Check section A for a detailed description.

## 3.2 Prompt-guided Flow Estimation

When predicting an optical flow map from a pair of input frames, the feature encoder extracts textural features to be matched pixel-wise during flow updates, while the context encoder focuses on semantic information to ensure the final flow outputs are meaningful and smooth within object-wise regions. Therefore, to achieve accurate flow estimation for specific ROIs, SAM2Flow conditions encoded context with foreground information.

**Prompt-conditioned Context**. Inspired by SAM, SAM2Flow takes user input prompts that specify ROIs that are are more diagnostically relevant, such as different vessels or branches of a complex vessel structure, to condition focused flow estimation from the context encoder, in cases where complex overlapping vessel structures or irrelevant background motion (e.g., heartbeat, tissue movement) can mislead outputs.

Our model takes several sets of points and corresponding labels $l_i \in \{1, 0\}$ (foreground-1 / background-0) as prompts $p_s = \{(x_i, y_i; l_i)\}_i^{N_p}$. Each point prompt set representing an ROI is encoded into a 1D prompt vector with a prompt encoder, $\mathcal{E}^p(p_s) \in \mathbb{R}^{1 \times D_p}$. Whenever there are user prompt inputs at a certain frame, the encoded context is augmented by the prompts via two-layer cross-attention:

$$\mathcal{P}(\mathcal{E}^c(I_t))^i = \text{Cross-Attention}[\mathcal{E}^c(I_t), \mathcal{E}^p(p_s^i)] \tag{1}$$

where $\mathcal{P}(\mathcal{E}^c(I_t))^i \in \mathbb{R}^{h \times w \times D_c}$ is the context prompted by the $i^{th}$ ROI point set. During flow estimation, the prompted context would guide the flow update module, $\mathcal{U}$, to focus on the refinement for the foregrounds and output a flow map for each region:

$$\Delta f^i = \mathcal{U}(\mathcal{P}(\mathcal{E}^c(I_t))^i, m_t^i) \tag{2}$$

If there is no user prompt input as in default, the model only predicts one unprompted flow map for the whole frame.

**ROI-guided Correlation Lookup**. One of the major computational bottlenecks in the current optical estimation model is the correlation search. $\mathcal{C}_k \in \mathbb{R}^{h \times w \times \frac{h}{2^k} \times \frac{h}{2^k}}$ is the $k^{th}$ layer of downsampled correlation volume in the pyramid. In RAFT-like architectures, at the beginning of each flow update iteration, the correlation map $Corr \in \mathbb{R}^{h \times w \times K}$ is generated by retrieving the correlation values for all pixels at all levels of the pyramid, based on the current flow prediction, resulting in a complexity of $O(hwK)$.

As flows in OBM videos are highly localized, occupying only a small portion of the entire field of view, SAM2Flow speeds up the `lookup` operation and suppresses the background noise by retrieving only values for foreground regions. To guide the correlation lookup, SAM2Flow incorporates an ROI decoder, similar to the mask decoder in SAM2, that makes a prediction for an ROI $R^i \in \mathbb{R}^{h \times w \times 1}$ based on each prompted context. The ROI decoder also outputs a 1D object pointer vector $\mathcal{O}^i \in \mathbb{R}^{1 \times D_\mathcal{M}}$, that is used for context memory in the following section:

$$R^i, \mathcal{O}^i = \mathcal{D}^R(\mathcal{P}(\mathcal{E}^c(I_t))^i) \tag{3}$$

Therefore, ROI-guided correlation lookups have a complexity of $O(N_R K)$, where $N_R$ is the total pixel number of ROIs.

## 3.3 Dual Memory Module

For robust performance over an extended video sequence (e.g., 12,000 frames for 60 seconds of OBM videos), SAM2Flow incorporates memory from previous time points to enhance current flow estimation. As optical flow is predicted by combining motion and context features, we propose the dual-memory mechanism, consisting of both motion and context memories. The motion memory ensures the long-term estimation smoothness for constantly pulsing flows, which is essential for downstream physiological analyses, while context memory helps to keep track of the identities of target vessels.

**Memory Encoding.** Previous flow predictions are encoded into motion memory and context memory by two separate memory encoders. We adopt memory encoders from SAM2 as our backbones. For motion memory at $t_0$, the memory encoder directly takes the flow map of each ROI as input and fuses it with the corresponding motion feature map:

$$\mathcal{M}_{t_0}^m = \mathcal{E}^{\mathcal{M}_m}(f_{t_0}, m_{t_0}) \in \mathbb{R}^{h \times w \times D_{\mathcal{M}}} \tag{4}$$

For context memory, the flow map will be binarized into the flow mask $\mathcal{B}(f_{t_0}) \in \mathbb{R}^{H \times W \times 1}$ before being fed into the context memory encoder, where it is combined with unprompted context by the context memory encoder:

$$\mathcal{M}_{t_0}^c = \mathcal{E}^{\mathcal{M}_c}(\mathcal{B}(f_{t_0}), \mathcal{E}^c(I_{t_0})) \in \mathbb{R}^{h \times w \times D_{\mathcal{M}}} \tag{5}$$

**Memory Bank.** The encoded motion and context memories, along with object pointers, are then stored in a FIFO queue, named the memory bank. To save space for long-video inference, the memory bank is limited to store memories up to $N$ recent frames. When the memory bank is full, the earliest memories are discarded. When there are memories from $n$ recent frames $(0 < n \le N)$ in the memory bank at a time point, $t_p$, the model retrieves all memories from the memory bank and stacks them into $\mathcal{M}_{t_p \sim n}^m, \mathcal{M}_{t_p \sim n}^c \in \mathbb{R}^{n \times h \times w \times D_{\mathcal{M}}}$.

**Motion Memory Attention.** The microvascular flow pattern within a specific ROI is usually temporally smooth and predictable. For example, the blood in a certain vessel tends to flow in the same direction throughout the whole video. Therefore, SAM2Flow introduces motion memory that stabilizes flow regression. We utilize a stack of *vanilla* attention blocks of alternating self- and cross-attention [50] to condition the current motion feature with motion memories:

$$\text{Mem}(m_{t_p}) = Att^m(m_{t_p}, \mathcal{M}_{t_p \sim n}^m) \tag{6}$$

**Context Memory Attention.** For reliable flow estimation over time, the context branch of SAM2Flow should provide stable semantic information and segmentation of ROIs. Context memory propagates user-defined ROIs across frames. As a result, SAM2Flow only requires sparse user inputs in the first few frames of a long video. To achieve this, we condition the current context from context encoder with the stacked context memory and object pointers, using the same attention operation as eq. (6):

$$\text{Mem}(\mathcal{E}^c(I_{t_p})) = Att^c(\mathcal{E}^c(I_{t_p}), \mathcal{M}_{t_p \sim n}^c, \mathcal{O}_{t_p \sim n}) \tag{7}$$

When it comes to a frame with user prompt inputs, SAM2Flow prioritizes the user prompts and skips context memory attention. With the memory-augmented motion and context features, the flow update module is able to generate temporally smooth flow estimations with the same iterative refinements as eq. (2) for each ROI:

$$\Delta f^i = \mathcal{U}(\text{Mem}(\mathcal{E}^c(I_{t_p})^i), \text{Mem}(m_t^i)) \tag{8}$$

## 4 Experiments

### 4.1 Datasets

**Microvascular Dataset.** We establish an *in-vivo* microvascular flow dataset that is larger than most of the existing public optical flow datasets, including 75 videos, with the paired ground truth flow maps of 306,800 frames in total. The grayscale videos are collected by imaging the superficial capillaries in the oral cavity of 15 healthy volunteers using the OBM system [6], at 200FPS with a frame size of $512 \times 512$. All participants gave written informed consent, and experiments were conducted under a Johns Hopkins University Institutional Review Board-approved protocol (IRB00264150). We split the dataset into training, validation, and testing subsets, containing 45, 15, and 15 videos, respectively. The flow map ground truths are generated using the spatiotemporal diagram [14] with manual refinement.

**Public Datasets.** We use two public datasets to test the generalization ability of SAM2Flow outside of the microscopic domain. Sintel [51] and Spring [52] are both popular optical flow datasets with long animation sequences and pixel-accurate flow ground truth. Since this paper proposes a novel challenge of joint motion ROI detection and optical flow estimation, we apply the SAM2 model to generate panoptic ROI masks on these two dataset. We utilize the Spring training dataset, consisting

Table 1: Comparative study of optical flow estimation performance on Microvascular test set.

| Model | Whole Image | | | | Foreground | | | | Speed |
|---|---|---|---|---|---|---|---|---|---|
| | EPE↓ | 1px↑ | 3px↑ | 5px↑ | FEPE↓ | 5px↑ | 10px↑ | 15px↑ | mspf↓ |
| RAFT [16] | 3.18 (2.61) | 0.86 | 0.89 | 0.91 | 27.73 (24.79) | 0.39 | 0.52 | 0.56 | 51.48 |
| GMA [17] | 3.22 (3.66) | 0.87 | 0.89 | 0.91 | 28.34 (26.82) | 0.38 | 0.54 | 0.58 | 43.66 |
| SEA-RAFT[46] | 1.28 (1.03) | **0.88** | 0.92 | 0.94 | 6.60 (5.47) | **0.69** | **0.86** | 0.91 | **21.14** |
| FlowFormer++ [44] | 1.72 (1.38) | **0.88** | 0.91 | 0.93 | 10.89 (9.28) | 0.60 | 0.78 | 0.84 | 133.95 |
| VideoFlow_BOF[(MF)] [18] | 3.28 (2.51) | 0.86 | 0.87 | 0.88 | 28.16 (26.64) | 0.15 | 0.32 | 0.41 | 112.67 |
| MemFlow[(MF)] [20] | 1.79 (1.40) | **0.88** | 0.91 | 0.93 | 12.47 (10.23) | 0.58 | 0.74 | 0.80 | 43.98 |
| StreamFlow[(MF)] [19] | 1.43 (1.02) | **0.88** | 0.90 | 0.93 | 10.13 (8.36) | 0.49 | 0.74 | 0.84 | 60.07 |
| **SAM2Flow**[(MF)] | **1.14 (0.92)** | **0.88** | **0.93** | **0.96** | **5.84 (4.86)** | 0.66 | **0.86** | **0.93** | 48.78 |

∗ [(MF)] indicates multi-frames optical flow models; best performance is **highlighted**, while second best performance is underlined; **EPE** & **FEPE**: Mean(Standard Deviation, SD); **mspf**: milliseconds per frame.

37 videos, 10,000 paired flow GTs, and then create training, validation, and test splits with 25, 5, and 7 videos, respectively. For Sintel dataset, we split the total of 23 scenes into training, validation, and test with 14, 3 and 6 videos. The Sintel training and validation combine both Clean and Final videos. For evaluation, we report the performance on the Clean and Final videos separately. More details about datasets, video pre-processing, and ground truth generation are described in section B.

### 4.2 Implementation

**SAM2Flow Settings.** All the models are trained and tested on two NVIDIA RTX A5500 GPUs. The feature and context encoders downscale inputs 8 times to 64x64. SAM2Flow runs 8 flow update iterations for training, and 4 for inference. We set the number of correlation pyramid layers $K$ to 4, and the memory bank limit $N$ to 7. For SAM2Flow training, we input an 8-frame video at each step with MoL loss from SEA-RAFT, using the Adam optimizer and global learning rate starting at 1e-4, with weight decay and scheduler. We use a multi-stage training strategy detailed in section C.1

**Comparative Experiments.** To demonstrate SAM2Flow's performance, we conduct the comparative study with six SOTA baseline models, including **Two-frame Models:** RAFT[16], GMA[17], SEA-RAFT[46], FlowFormer++[44], and **Multi-frame Models:** VideoFlow[18], MemFlow[20], and StreamFlow[19] on both Microvascular and Spring datasets. For fair comparison, we empirically choose the pretrained checkpoints and fine-tune the model on the corresponding datasets following the configurations from the papers. (Details at section C.2.)

**Evaluation Metrics.** For microvascular videos, we evaluate accuracy with end-point errors (EPE) and percentages of pixel errors within [1, 3, 5px] for the whole frames. Since the SAM2Flow focuses on the informative ROIs, we also report foreground EPE (FEPE) and larger pixel error rates [5, 10, 15px] within foregrounds, due to the rapid blood flow ($\sim$ 30px/frame), shown in fig. 2. For microvascular videos, foregrounds are defined as in-focus vessels with active blood flow. For Sintel video, we also report FEPEs on both Clean and Final sets. Following previous works on Spring dataset, we report FEPE, foreground 1px errors, foreground flow outlier rate (Fl) that is defined as > 3pxs and > 5% of GT flow magnitude), the average EPE of flow outliers (Fl-epe), as well as foreground weighted AUC (WAUC). The foreground ROIs in Sintel and Spring videos are defined as the objects or regions with motions that are distinctive from the background scenes.

## 5 Results

### 5.1 Microvascular Flow Prediction Performance

The baseline models and SAM2Flow are evaluated on the Microvascular test split. The quantitative results are illustrated in table 1. The best single-frame baseline performance comes from SEA-RAFT, achieving an EPE $= 1.28(SD = 1.03)$ and a foreground EPE $= 6.60(5.47)$, with the fastest inference speed among all models ($21.14ms$), thanks to the light-weighted encoders and fewer flow update iterations. FlowFormer++ also achieves strong performance FEPE $= 10.89(9.28)$ but at a much higher computational cost ($133.95ms$) due to its large transformer-based cost-volume encoder.

Table 2: Comparative study of foreground optical flow estimation on Sintel and Spring datasets.

| Model | Sintel-FEPE | | Spring-Foreground | | | | |
|---|---|---|---|---|---|---|---|
| | Clean↓ | Final↓ | FEPE↓ | 1px (%)↑ | Fl (%)↓ | Fl-epe↓ | WAUC↑ |
| RAFT [16] | 5.21 (9.30) | 5.47 (10.12) | 2.25 (6.97) | 74.25 | 9.87 | 7.98 (8.98) | 74.76 |
| GMA [17] | 4.65 (7.45) | 5.14 (8.17) | 2.17 (6.30) | 76.21 | 9.08 | 7.67 (7.72) | 79.67 |
| SEA-RAFT[46] | 3.26 (7.54) | 4.08 (8.94) | 1.45 (5.61) | 86.32 | **5.18** | 8.18 (10.85) | 83.85 |
| MemFlow[(MF)] [20] | 3.77 (5.82) | 4.27 (7.09) | 1.56 (7.27) | 86.49 | 7.24 | 8.82 (12.72) | 83.54 |
| StreamFlow[(MF)] [19] | 4.06 (**5.37**) | 4.43 (**5.66**) | 1.54 (7.23) | 85.16 | 5.82 | 7.74 (10.16) | 82.51 |
| **SAM2Flow**[(MF)] | **3.17** (6.89) | **3.39** (5.97) | **1.23 (4.21)** | **87.13** | 5.29 | **7.30 (5.29)** | **84.57** |

Multi-frame models generally infer more slowly than single-frame approaches as models process additional temporal information. MemFlow achieves reasonable flow estimation accuracy $\text{FEPE} = 12.47(10.23)$ with a fast inference speed ($43.98s$), with only one frame of memory[20]. SAM2Flow outperforms all other models in whole-image $\text{EPE} = 1.14(0.92)$ and $\text{FEPE} = 5.84(4.86)$, indicating its superior flow estimation accuracy. Compared to single-frame models and MemFlow, SAM2Flow maintains a competitive inference speed ($48.78ms$) while incorporating long-term dual memories of 7 frames. **Overall, these results highlight the effectiveness of SAM2Flow in balancing high accuracy with efficient computation, outperforming both single-frame and other multi-frame models in critical performance metrics.**

We note that our test set ground truths do not have pixel-level accuracy, as they are derived from ST diagrams, a spatially and temporally smoothed estimation. As well-trained models achieve lower errors, quantitative comparisons with the current ground truth may become less indicative of true performance. Consequently, in section 5.6, we conduct a more in-depth analysis to assess whether the predicted flow maps accurately reflect the actual flow patterns from the videos.

## 5.2 Public Benchmark: Sintel

On the Sintel benchmark, SAM2Flow demonstrates superior performance in foreground flow estimation compared to all baselines. As shown in table 2, SAM2Flow achieves the lowest errors on both Clean ($\text{FEPE} = 3.17$) and Final ($\text{FEPE} = 3.39$) videos. While SEA-RAFT ($\text{FEPE} = 3.26$) and MemFlow ($\text{FEPE} = 3.77$) show good accuracy in Clean videos that contains simpler scenes with less texture, their performances deteriorate when adding more textures and illumination effects to the scene, as in Final videos (SEA-RAFT $\Delta\text{FEPE} = 0.82$ and MemFlow $\Delta\text{FEPE} = 0.5$); on the contrary, SAM2Flow shows strong robustness ($\Delta\text{FEPE} = 0.22$) to these challenges for optical flow estimation within the motion ROIs.

## 5.3 Public Benchmark: Spring

The quantitative results on Springs are shown in the second half of table 2. For foreground flow estimation, SAM2Flow demonstrates the strongest performance $\text{FEPE} = 1.23(4.21)$, with a 15% improvement compared to the best baseline model, SEA-RAFT $\text{FEPE} = 1.45(5.61)$. With the context-guided focus on motion foregrounds, SAM2Flow achieves finer estimation, especially for fast and complex motions within the ROIs, resulting in low flow outliers: $\text{Fl} = 5.29\%$, $\text{Fl-epe} = 7.30$. The visual comparison can be found in the fig. 3. These results, combined with the strong performance on the Sintel dataset, highlight that **SAM2Flow achieves impressive generalizability and the state-of-the-art ROI-centric optical flow estimation beyond the microscopic domain.**

## 5.4 Ablation Study

The results of the ablation study are shown in table 3. We conduct ablation studies to test the efficacy of memory modules and ROI-guided lookup operation to flow estimation accuracy and efficiency.

**Backbone Encoders.** *Backbones-Only* is a simple combination of pre-trained SAM2 context encoder and the SEA-RAFT feature encoder backbone with limited fine-tuning on our dataset. Even though $\text{FEPE} = 9.75$ of this basic setup is among the SOTA performance, it does not outperform the original SEA-RAFT model (table 1), indicating that this backbone configuration has the potential to be further improved by leveraging public datasets.

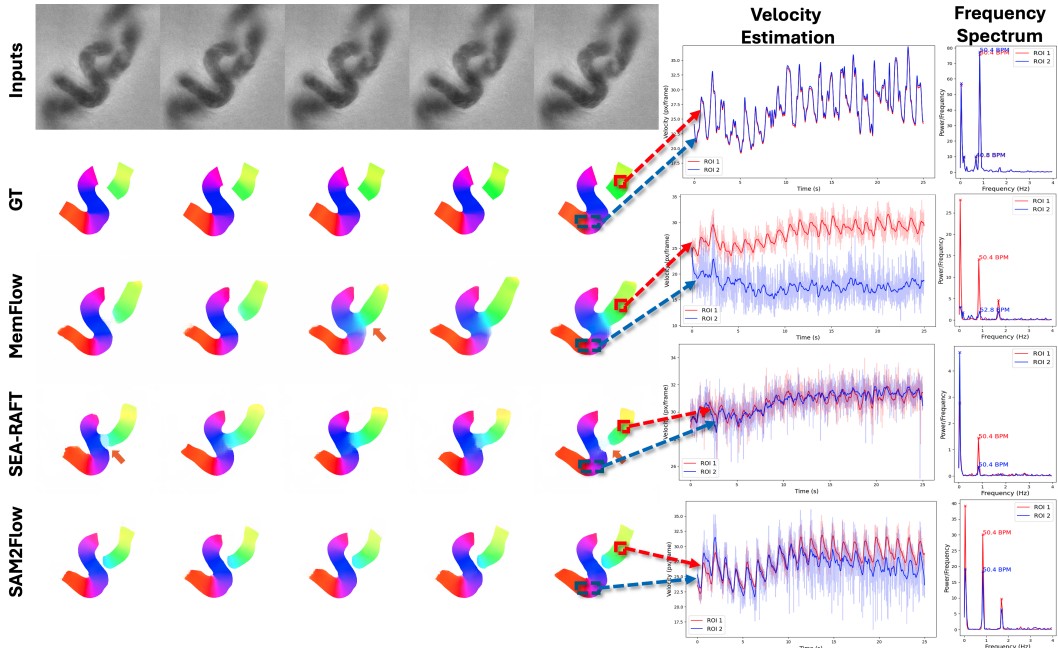

Figure 2: **Qualitative comparison on the microvascular test video. Left:** Five representative frames, GT flow maps, and predictions from TOP3 models, on a challenging video where two blood flows cross. Arrows highlight regions where MemFlow and SEA-RAFT mistakenly merge the two streams, while SAM2Flow maintains distinct ROIs throughout the sequence. **Right:** Velocity estimations at two fixed ROIs (dashed boxes) with corresponding frequency spectra over 5000 frames. SAM2Flow generates the most robust and clinically meaningful velocity estimations from both flows.

**Dual Memory Mechanism**. We tested the motion and context modules separately. *M-Mem Only* adds a motion memory module to the backbones, while *C-Mem Only* contains only the context memory module. Based on the Foreground accuracy results, the context memory module better boosts the flow estimation accuracy ($\Delta$FEPE $= -3.43$), as the context encoder from the SAM ViT backbone is tailored to work with the memory mechanism. On the contrary, motion features encoded by the CNN-based encoders benefit less from the memory mechanism ($\Delta$FEPE $= -2.34$). Meanwhile, motion memory yields a heavier computational overhead ($\Delta T = 16.22ms$) as its attention is embedded within the iterative flow updates. *No ROI* consists of the complete dual memory module and encoder backbones. As the results suggest, the dual memory module outperforms either of the context/motion-only memories, achieving the best FEPE of all experiments. Therefore, the final SAM2Flow integrates both context and motion memories to achieve the most robust flow estimation.

Table 3: Ablation study on Microvascular validation set.

| Method | Foreground | | | | Speed |
|---|---|---|---|---|---|
| | FEPE↓ | 5px↑ | 10px↑ | 15px↑ | mspf↓ |
| Backbones | 9.75 | 0.54 | 0.77 | 0.83 | **32.76** |
| M-Mem Only | 7.41 | 0.63 | 0.81 | 0.88 | 48.98 |
| C-Mem Only | 6.32 | 0.64 | 0.84 | 0.91 | 38.63 |
| No ROI | **5.12** | 0.69 | **0.89** | **0.94** | 55.25 |
| **SAM2Flow** | 5.46 | **0.72** | 0.87 | 0.92 | 48.78 |

\* M-Mem: Motion memory; C-Mem: Context memory.

**ROI-guided Lookup**. *No ROI* does not include ROI-guided correlation lookup. To achieve higher efficiency, the final SAM2Flow uses ROI segmentation to guide correlation lookup, avoiding indexing irrelevant background pixels. It speeds up inference by $12\%$ without significantly affecting accuracy.

### 5.5 Qualitative Comparison

We visualize and compare SAM2Flow flow estimates with the best-performing single-frame model, SEA-RAFT, and multi-frame model, MemFlow, on sample frames in fig. 2. The sample video is a challenging case, as two separate flows cross each other. The results suggest that both MemFlow and

SEA-RAFT get confused and merge two flows by mistake. On the other hand, with the help of sparse prompts in the first few frames, SAM2Flow outputs robust flow estimations as two separate flows. **The visualization suggests that SAM2Flow achieves stronger flow estimation in more complex flow structures**, especially when flows are closely tangled in the FoV.

### 5.6 Physiological Applications

We analyzed velocity estimation from the Top3 models throughout 5000 frames ($25s$) on the sample test video. Two velocity-over-time plots are generated by averaging the flow estimation within the fixed subregions within two vessels. We also plot the frequency spectra to verify the extracted pulsatile patterns. SAM2Flow yields the strongest heart rate signals (50.4 BPM) from both flows. With temporal memory, SAM2Flow is less susceptible to noise. Taking a closer look at the waveforms, the pattern within each cardiac cycle from SAM2Flow is closest to a meaningful clinical waveform from other means of measurement [7, 53], including a central peak (peak systolic velocity), a subpeak, and a central trough (end-diastolic velocity). More clinically relevant biomarkers, such as peak ratio and resistive index [53], could be characterized based on SAM2Flow velocity estimation. **Across long videos, SAM2Flow achieves more robust and accurate velocity estimation.**

### 5.7 Limitations

Due to the focus on foreground estimation and ROI-guided correlation lookup, SAM2Flow relies on robust ROI detection from the context branch. Therefore, the flow estimation would deteriorate due to failed or incomplete ROI detection in some complex scenes. However, our model design mitigates the effect on flow estimation from different types of ROI errors:

**Over segmentation.** When ROI is overly segmented, some background pixels are also indexed in correlation lookup. However, this wouldn't directly degrade flow estimations, as the trained motion encoder and RNN further delineate the motion boundary through iterative flow updates. Full-frame flow regression could be considered as an extreme case of over-segmentation.

**Transient frames drop or incomplete ROI.** Motion memory module provides redundancy against occasional failures by maintaining consistency with prior memories (section 3.3). This feature stabilizes predictions when one frame has incomplete or no ROI detection. This is demonstrated by the ablation study of the motion memory module, where FEPE improves from $6.32$ to $5.46$ (table 3). SAM2Flow incorporates an additional trick like warm start (section A), to further stabilize the flow estimation in cases of occasional ROI instability.

**Missed/incomplete ROI over an extended period.** Failed ROI detection over more consecutive frames would result in failed optical flow estimation, since the model fails to identify any target. However, as an interactive pipeline, SAM2Flow offers the flexibility to add/correct user prompts for these challenging frames during inference. We recommend using multiple positive points spaced over the ROI for larger targets to mitigate prompt ambiguity.

## 6 Conclusion

In this paper, we first identified a new challenge of ROI-centric optical flow estimation over long sequences. Therefore, we introduce SAM2Flow, an interactive optical flow estimation model for *in vivo* microvascular flow from OBM videos. Our technique enables user-specified ROIs through sparse point prompts for ROI-specific flow estimation. To ensure that flows remain temporally consistent, even in fast and optically ambiguous data, we propose a novel dual memory attention mechanism, comprising both motion and context memory. We demonstrate the effectiveness of SAM2Flow by testing against 6 other baseline models on both microvascular videos. The proposed model achieves the best EPE and foreground EPE among all baselines on the test set. Additional experiment on the public Spring dataset indicates the promising generalizability of SAM2Flow beyond microscopic videos. For future work, we aim to further boost model speed and flow estimation accuracy by exploring more compact encoder backbones, such as SAM2-B+ and SAM2-S, and on diverse datasets. Moreover, we would adapt SAM2Flow to microvascular flow with various conditions, such as sickle cell and sepsis patient data. In summary, SAM2Flow introduces innovative techniques to address the unique challenge of region-specific optical flow. This approach empowers physicians to extract microcirculation biomarkers from OBM video data non-invasively.

**Acknowledgement.** This work was supported in part by research funding from the Gates Foundation (INV-006005).

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

## Technical Appendices and Supplementary Material

## A  Flow Initialization

**ROI Registration.** For *in vivo* microscopic videos, subject movement and camera jitter are inevitable. Large movements pose a significant challenge to optical flow estimation. Therefore, during inference time, we introduce an ROI registration step to initialize the flow map with a global offset value. In OBM videos, the background movements are mostly 2D translation, occasionally with slight rotation. To keep ROI registration most efficient, we ignore the rotation and apply the fastest strategy by calculating the center displacement of masks from the ROI decoder. This simple strategy helps remove large background movements without extensive computational overhead. The related ablation study is included at section D.2

**Warm Start.** As mentioned in section 3.3, the flow patterns within the video are temporally smooth. In addition to the motion memory, we use the flow map prediction from the last frame to initialize the current flow estimation, which has been proven to facilitate flow regression convergence in previous OF models [16]. For diagram simplicity, this step is not shown in fig. 1.

## B  Dataset

### B.1  Data Collection

The dataset consists of 75 videos from 15 healthy volunteers. The 20x OBM system is applied to superficial capillaries inside the oral cavity. Videos were recorded at 200 FPS to ensure less motion blur of fast-flowing blood cells. Each raw video is recorded for at least 90 seconds. Videos are manually reviewed and the most in-focus and stable segments are selected for further processing.

### B.2  Dataset Preparation

**Video Processing.** To generate an optical flow map ground truth, OBM videos first go through a series of preprocessing steps in ImageJ. 1) Background corrections: a temporally averaged frame over the video is calculated first, then a Gaussian blur with a kernel size of 100. The original frames are flattened by dividing by the background estimation. Finally, frames are normalized to [0, 255]. 2) Video stabilization: We use template matching to align slices in the stack with the normalized correlation coefficient.

**Flow Estimation.** 1) Flow mask: with the stabilized video frames, the flow mask is calculated from the standard deviation over time. The mask is then binarized into flow masks using Otsu thresholding. When there is more than one flow, multiple binary masks are generated. Output masks are further refined by human efforts. 2) Centerline detection: With the refined flow masks, the centerline of each flow is generated by flow mask skeletonization. Small branches and ends are removed so that only the main branches are used for flow estimation. 3) ST diagram: Based on the centerline, the intensity profile along the centerline over time is plotted across the whole frame. Parallel lines around centerlines are also used to generate ST diagrams. 4) Flow velocity estimation: from ST diagrams, the angle to tilted lines is calculated by Hough line transform using a sliding time window of 235 frames.

**Optical Flow Map Generations.** 1) Temporal smoothing: to remove the local noises from the ST diagram, the estimated velocity over time is smoothed by a Butterworth low-pass filter; 2) Velocity profile: velocity profiles are generated along the centerline from the above estimation. We also generate velocity profiles from nearby lines within the flow that are parallel to the centerline. 3) Velocity interpolation: after getting the velocity profile along multiple lines within the flow, the velocity map is generated for each frame by cubic interpolation. 4) Flow direction: The general direction of the flow (upward/downward) is manually decided by scrolling through the video. The local direction is determined by the tangent angle of the centerline curve. 5) Optical flow map: The final optical flow ground truth is generated by multiplying the velocity map with the direction map.

The final dataset is made up of 75 long videos paired with 306,800 ground truth flow maps, which is significantly longer and larger than most of the existing public datasets, as shown in table 4.

### B.3 Public Benchmark: Sintel and Spring

To evaluate SAM2Flow's ability to generalize beyond the microscopic domain, we benchmark it against baseline models on the public Sintel [51] and Spring [52] datasets. As the proposed task focuses on ROI optical flow estimation, standard whole-frame leaderboard metrics are not informative enough about the model's performance. Therefore, we apply custom splits on the Sintel training set and Spring training set and introduce additional ROI ground truths. The

Table 4: Dataset Size Comparison

| Dataset | Videos # | Flowmaps # | Avg video length (flowmaps/video) |
|---|---|---|---|
| Sintel[51] | 23 | 1,041 | 45 |
| Spring[52] | 37 | 10,000 | 270 |
| KITTI 2015[54] | 400 | 1,600 | 4 |
| FlyingChairs[55] | N/A | 22,872 | N/A |
| **Microvascular** | 75 | 306,800 | 4,091 |

Sintel frames are resized to $436 \times 960 \times 3$, and the Spring frames are downsampled by $2\times$ to $540 \times 960 \times 3$. The ROI masks are generated via panoptic segmentation with the pretrained SAM2-L model, prompted by $16 \times 16$ grid points over the whole frame. For evaluation, we only retain the segmented objects or regions with motions that are clearly distinct from the background/camera movement.

## C  Additional Implementation Details

### C.1  SAM2Flow Multi-stage Training

**Stage 1:** Finetune backbones on whole frame for domain adaptation. For backbones, we select the SEA-RAFT checkpoint pretrained on a mix of five optical flow datasets, including TartanAir[56], Sintel[51], FlyingChairs[55], KITTI[54] and HD1K[57]. We select the pretrained SAM2-L[22] checkpoint as the context backbone. Since the backbone SAM2 and SEA-RAFT have no prior information of the OBM domain, we train the two backbones separately on their original tasks for domain adaptation. The SEA-RAFT backbone is initially trained with whole-frame flow ground truth for 50,000 steps, with a $1e-4$ learning rate. And the SAM2 adapter is trained with the frozen SAM2 backbone on segmentation masks of flows across videos for 50,000 steps, with a $1e-4$ learning rate.

**Stage 2:** Module fusion. The trained SEA-RAFT and SAM2 encoder weights from stage 1 are imported into the SAM2Flow model. We disable the memory module at this stage and train the SAM2Flow on image pairs. We randomly generate 3-6 positive and negative point prompts based on the mask ground truths. The training is supervised on the masked optical flow ground truths. We used a smaller learning rate for the SAM2Flow weights at $5e-5$, except for the frozen SAM2 encoder.

**Stage 3:** Training with memory. As the last stage of training, we enable context and motion memory modules and input an 8-frame sequence for flow prediction. The point prompts are randomly generated for the first 1-3 frames, with 3-6 points for each frame. During training, we start the general learning rate at $1e-4$, freeze the weights of the SAM2 encoder, and use a learning rate scaling factor of 0.2 for the pre-trained SEA-RAFT feature encoder.

### C.2  Comparative Experiments

We elect to use weights from the most comprehensive pre-training checkpoints from each optical flow model. 1) **Two-frame baseline models**: We take pretrained weights on the mix of five datasets (same as section C.1) for SEA-RAFT, Sintel weights for RAFT, GMA, and FlowFormer++. The pretrained models are fine-tuned according to the training plans suggested by their respective open-source implementations [16, 17, 44, 46]. Models were trained on a workstation containing 2 NVIDIA RTX A5500 GPUs. Default configurations were used for each model. 2) **Multi-frame baseline models**: Two multi-frame baseline models are tested, VideoFlow and MemFlow [18, 20] pretrained Sintel. VideoFlow was trained using the provided three-frame training methodology due to the GPU memory. The default configuration was used for finetuning on our dataset, with the image size adjusted to fit our dataset. We ensure that there are enough steps so that all the models converge on the validation metrics during training on the Microvascular dataset and the Spring dataset.

# D  Additional Results

## D.1  Public Benchmarking Results

The visual comparison on a representative video segment from the Spring test split is shown in fig. 3. The qualitative error maps verify the quantitative gain. The context-guided foreground detection enables SAM2Flow to focus on tracking complex ROI motion with sharp flow edges. In contrast, MemFlow and SEA-RAFT exhibit poor performance and boundary leakage in the highlighted regions.

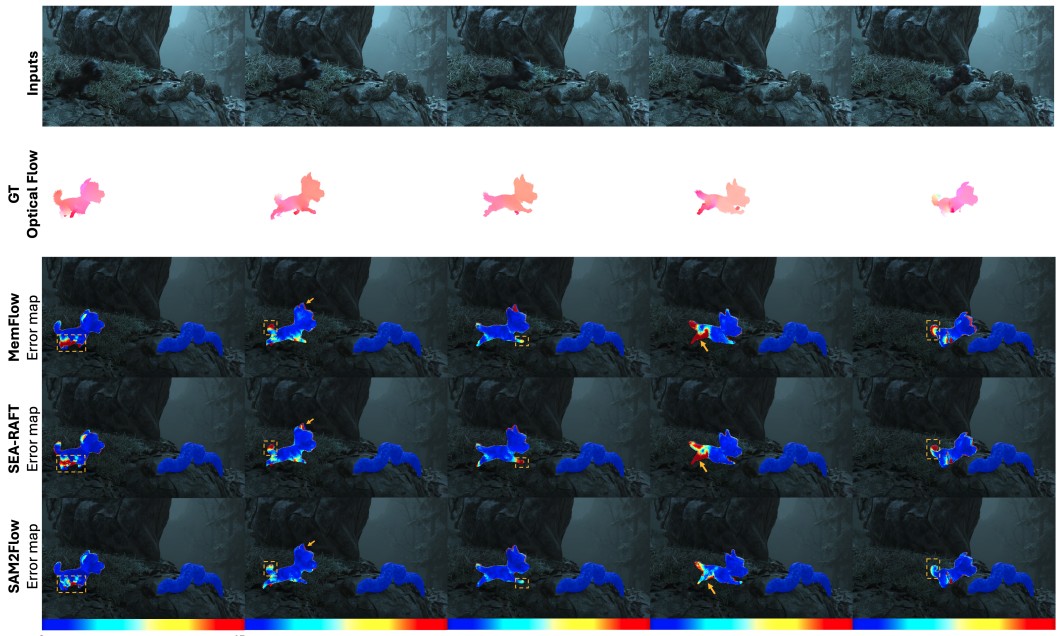

Figure 3: **Qualitative comparison on the Spring public benchmark.** Top: five consecutive animation frames from test video. Second row: foreground GT optical flow fields. Rows 3 − 5: Absolute foreground endpoint-error heatmaps (0 − 15 px, blue → red) overlay on the frames, for MemFlow, SEA-RAFT, and the proposed SAM2Flow. All the models perform well in estimating the optical flow of the object on the right side of the frame with little motion. Yellow arrows and dashed boxes highlight regions where the baselines yield higher errors on fast, complex ROI motions. Meanwhile, SAM2Flow produces robust and accurate flow estimation while preserving motion boundaries across the sequence.

## D.2  Ablation Study

Table 5: Ablation study for flow initialization.

| Method | Dataset | FEPE↓ | 5px↑ | 10px↑ | 15px↑ |
|--------|---------|-------|------|-------|-------|
| No FI | unstable | 20.50 | 0.37 | 0.52 | 0.66 |
| FI | unstable | 12.32 | 0.54 | 0.65 | 0.78 |
| No FI | stable | 5.84 | 0.66 | 0.86 | 0.93 |

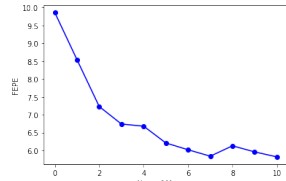

Figure 4: FEPE vs. number of memories.

**Flow initialization for unstable videos.** We proposed ROI registration as flow initialization (FI) at inference time, table 5 shows the ablation study results on unstable videos without retraining the model. ROI registration as flow initialization effectively improves the estimation accuracy on the

unstable videos, with a $40\%$ improvement in FEPE without any additional model adaptation on the background movements.

**Number of Memory vs. FEPE.** We test the effect of numbers of memory frames on the foreground EPE. The results are shown in fig. 4. We only test for the combined context and motion memory, from 0 to 10 frames. As the plot shown, FEPE first decreases and then converges around 7. SAM2Flow memory module performance exhibits marginal gains by enlarging memory banks beyond 7 frames, with linear compute memory growth. Therefore, we choose 7 frames as memory bank limit for the final SAM2Flow configuration.

