# OpenReview forum: "SAM2Flow: Interactive Optical Flow Estimation with Dual Memory for in vivo Microcirculation Analysis"
_NeurIPS.cc/2025/Conference — NeurIPS 2025 poster_

### Official Review · Reviewer_D1JP · 2025-06-16

**Clarity:** 3
**Significance:** 3
**Originality:** 4
**Rating:** 4
**Confidence:** 3

**Summary:**

This paper proposed a SAM-enhanced model to estimate optical flows of blood in microcirculation analysis. The key contributions are the dataset and dual-memory modules (motion and context memories). On one in-house dataset and another public dataset, the performance is good to estimate the optical flows.

**Questions:**

1. Please include the complexity comparisons in the tables.

2. How is the performance when using bboxes as prompts? Here, the point prompts are a bit time-consuming.

3. Discussion of failure cases

**Ethical Concerns:**

["NO or VERY MINOR ethics concerns only"]

**Final Justification:**

The authors have addressed my concerns. I keep my score as Borderline accept.

**Limitations:**

There is a limitation discussion about the imperfect prompts in the model. No failure cases.

**Quality:**

3

**Strengths And Weaknesses:**

Strengths:

1. The studied problem is important, and sharing the dataset is a good contribution.

2. Extensive experiments have shown that the model achieves good performance.


Weaknesses:

1. The NO-ROI setting can achieve better performance in Table 3. More justifications are needed.

2. [50] can achieve comparable and even better performance. The authors are suggested to clearly discuss its superiority and contributions.

3. More prompts can be explored to validate its generalizability.

4. The authors are suggested to analyze failure cases, especially for some subtle objects which cannot be segmented well by SAM2.

---

> ### Author Rebuttal · Authors · 2025-07-30
>
> We would like to thank the reviewer for recognizing the significance of the in-vivo blood flow estimation task and the contribution of our microvascular dataset, as well as the strong performance of the proposed SAM2Flow model. Below, we provide detailed responses to your suggestions and comments.
>
> ## 1. Justification of “No-ROI” Performance in Table 3
> **The "No ROI" setting in Table 3** refers to a variant where ROI-guided correlation lookup is disabled (**Line 328**), meaning all pixels are considered during flow regression (i.e., full-frame regression). For the complete SAM2Flow framework, we introduce the ROI-guided lookup to guide correlation search within the detected ROI and suppress the computation on the irrelevant background.  As shown in **Table 3**, ROI guidance enhances model efficiency (**inference time improves from 55.25 ms to 48.78 ms**) at the cost of some raw accuracy (**FEPE = 5.46 vs. 5.12**). This trade-off is mainly from imperfect ROI segmentation, particularly false negatives, in some challenging scenarios.
>
> For the high-frame-rate microvasculation videos (12000 frames for one-minute video), we choose to prioritize the model efficiency, given the comparable estimation accuracy. ***We will clarify this trade-off more explicitly in Sec. 5.3 for the final version.*** And we also include a detailed discussion about the flow estimation under imperfect ROI detection in the next section.
>
> ## 2. Failure Case Analysis
> We agree with the suggestion that an expanded failure case analysis could provide a more comprehensive overview of the model performance, especially under challenging scenarios. As mentioned above and in the **limitations (Sec. 5.6)**, the flow estimation accuracy may be affected by imperfect ROI detections. Here, we provide a detailed analysis of how SAM2Flow copes with different types of ROI errors and the resulting flow estimations:
> * **Over segmentation.** When an over-segmented ROI guides correlation lookup, some background pixels are also indexed and fed to subsequent steps. However, this doesn’t degrade final flow estimations, as the trained motion encoder and RNN further delineate the motion boundary through iterative flow updates. Full-frame flow regression could be considered as an extreme case of over-segmentation.
> * **Transient frames drop or incomplete ROI.** Motion memory provides redundancy against occasional failures by retrieving memories from prior frames (**Sec. 3.3**) that stabilize predictions even if one frame has incomplete or no ROI detection. This is demonstrated by the ablation study of the motion memory module, where **FEPE dropped from 6.32 to 5.46 (Table 3)**. SAM2Flow incorporates an additional trick, known as warm start (**Supplement A**), to further stabilize the flow estimation in cases of occasional ROI instability.
> * **Missed/incomplete ROI over an extended period.** Failed ROI detection over more consecutive frames would result in failed optical flow estimation, since the model fails to identify any target. However, as an interactive pipeline, SAM2Flow offers the flexibility to add/correct user prompts for these challenging frames during inference. We recommend using multiple positive points spaced over the ROI for larger targets to mitigate prompt ambiguity.
>
> ***For the final version, we plan to include the above details and explicit visualization of failure cases for more intuitive discussion of limitations (Sec. 5.6).*** These additions will clarify where SAM2Flow still struggles and motivate future improvements in segmentation under ambiguous conditions.
> ## 3. Prompt Modality: Bounding Boxes vs Point Prompts
> For the microvascular videos, we chose point prompts to guide the flow localization in the context branch of SAM2Flow. Based on our observation, positive/negative point prompts can most effectively define any target vessels from single-vessel to complex multi-vessel scenes. Whereas bounding box prompts for different target ROIs would be ambiguous due to significant overlaps when multiple flows are close together and cross each other. For example, see **image $I_t$ in Fig 1**, which has two different vessels close to each other, and **Fig. 2**, which has two different entangled vessels.  Moreover, SAM2Flow identifies ROIs from temporally sparse prompts. Empirically, we only need prompts up to 5 frames for a one-minute-long (12,000-frame) microvascular OBM video. Therefore, the difference in time cost and manual labor is trivial.
>
> However, other forms of prompts could be more informative than point prompts when applying SAM2Flow to different videos. Thanks to the flexibility of SAM2's prompt encoder, **SAM2Flow can easily support various prompt inputs, including points, bounding boxes, and masks.**
>
> ## 4. Comparison with SEA-RAFT [50]
> SEA-RAFT is a strong two-frame model and serves as our motion encoder backbone. In our experiments, SEA-RAFT achieved competitive performance. However, we want to note that SAM2Flow outperforms SEA-RAFT on both datasets (**FEPE = 5.84 vs. 6.60 on the microvascular dataset, FEPE = 1.23 vs. 1.45 on the Spring dataset**). We have also conducted an additional comparative study on the **Sintel** benchmark (Please find details at **Q3 in Rebuttal to Reviewer 33dS**), where SAM2Flow achieves better foreground estimation accuracy in both “Clean” and “Final” videos (**FEPE-Clean=3.17 vs. 3.26; FEPE-Final=3.39 vs. 4.08**).
> | Model | FEPE in Sintel-Clean | FEPE in Sintel-Final |
> |:---:|:---:|:---:|
> | SEA-RAFT | 3.26 (7.54) | 4.08 (8.94) |
> | SAM2Flow | **3.17 (6.89)** | **3.39 (5.97)** |
>
> As a two-frame model, SEA-RAFT achieves faster inference (**21.14 ms/frame vs. 48.78 ms/frame** on 512x512 inputs). However, SAM2Flow, inherently a multi-frame model, achieves improvements in practical application and supports unique features:
> * Interactive ROI prompting;
> * Long-term temporal consistency via dual memory;
> * Robustness in multi-object and overlapping flow scenarios.
>
> These design features are not present in SEA-RAFT and are essential for the physiological analysis of long clinical videos with complex vascular flows, as shown in **Sec. 5.5** and **Fig. 2**.
> ## 5. Complexity metrics
> Thank you for pointing this out. The current **Table 1** reports **runtime (mspf)** as a measurement of model efficiency. ***We will include additional complexity columns (“number of FLOPs” and “memory use per batch”) in the updated Table 1 for the final draft.***

---

> > ### Comment · Reviewer_D1JP · 2025-08-05
> >
> > Thanks for the rebuttal. I believe the dataset could be a good contribution and keep my score as BA.

---

### Official Review · Reviewer_5zif · 2025-06-16

**Clarity:** 3
**Significance:** 3
**Originality:** 4
**Rating:** 5
**Confidence:** 4

**Summary:**

This paper introduces SAM2Flow, an interactive optical flow estimation model designed for analyzing long in vivo microvascular flow videos captured by Oblique Back-illumination Microscopy (OBM). Inspired by the Segment Anything Model (SAM2), the model enables users to specify regions of interest (ROIs) through prompts, focusing flow estimation on targeted vascular segments. A key innovation is the dual memory attention mechanism, combining motion and context memory, which ensures stable and efficient flow estimation over extended video sequences. SAM2Flow integrates a feature encoder from SEA-RAFT and a context encoder from SAM2, adapted via an adapter to the OBM domain, and employs ROI-guided correlation lookup to optimize computational efficiency. The model is designed to address the limitations of existing methods in handling long sequences and complex vascular structures. Experimental results demonstrate the effectiveness of the proposed method.

**Questions:**

[Questions]

Overall, the method proposed in this paper is novel, but there are still some confusing minor issues:

1. How reliable is the ROI detection in the proposed method?
2. When ROI detection is not fully reliable, how does it affect the estimation accuracy of the proposed method?
3. For the Spring dataset, how are foreground regions defined and generated? Could you provide a more detailed explanation?
4. In the studied scenarios, vascular overlapping in images seems to exist. How does the proposed method perform in such scenarios? Can the proposed strategies handle such scenarios effectively?
5. Open discussion: The paper mentions multiple microvascular imaging techniques, but it mainly focuses on OBM. Does the proposed method have generalizability to other microvascular imaging techniques? If its generalizability is limited, what are the reasons?

**Ethical Concerns:**

["NO or VERY MINOR ethics concerns only"]

**Final Justification:**

Thank you for the detailed responses from the authors. All my questions have been addressed, and I have decided to upgrade my rating to 5. Additionally, there are currently some works on optical flow based on different sensors [1-4]. Citing these works may help make the related work section more comprehensive.

Reference
[1] Z. Ding, et al., Spatio-Temporal Recurrent Networks for Event-Based Optical Flow Estimation. AAAI 2022.
[2] A. Luo, et al., Efficient Meshflow and Optical Flow Estimation from Event Cameras. CVPR 2024.
[3] R. Zhao, et al., Learning Optical Flow From Continuous Spike Streams. NeurIPS 2022.
[4] L. Xia, et al., Unsupervised Optical Flow Estimation with Dynamic Timing Representation for Spike Camera. NeurIPS 2023.

**Limitations:**

yes

**Quality:**

4

**Strengths And Weaknesses:**

[Strengths]

1. This paper proposes a specialized dataset tailored for optical flow estimation in in vivo microvascular studies, comprising 75 long-form videos and 306,800 paired ground-truth flow. The dataset can support downstream research in this area.
2. This paper Introduces interactive ROI-guided flow estimation via user prompts, enhancing clinical efficiency by focusing on relevant vascular regions and reducing unnecessary calculations.
3. The proposed methods employs a dual memory module with motion and context memories, improving long-sequence flow estimation stability for sustained microvascular dynamics.


[Weaknesses]

The primary limitation of the proposed method is its dependency on the context branch for ROI (region of interest) detection to guide flow estimation. In complex scenarios with intersecting vascular structures or turbulent blood flow, failed or incomplete ROI detection can directly degrade the accuracy of flow estimation.

---

> ### Author Rebuttal · Authors · 2025-07-26
>
> We sincerely thank the reviewer for the insightful feedback. We are pleased that you find our dataset, interactive ROI-guided flow, and dual-memory mechanism innovative and valuable for clinical and research applications. We appreciate your suggestions to clarify the model’s dependency on ROI detection.
> ## 1. How reliable is the ROI detection of SAM2Flow?
> Our ROI detection in the context branch is built upon the SAM2 encoder, which is known for its strong semantic generalization, and further fine-tuned for the microscopic flow domain via a lightweight adapter. Two features of SAM2, user-prompt and context memory, ensure superior ROI segmentation compared to other backbones, especially in terms of temporal stability. As shown in our **ablation study (Table 3)**, the context memory significantly improves estimation accuracy (**FEPE dropped from 9.75 to 6.32**), demonstrating reliable ROI detection and context encoding, which leads to more robust motion estimation over long sequences.
> ## 2. How does incomplete ROI detection affect estimation accuracy?
> ROI detection from the context branch is used to specify the target motion region and guide the correlation lookup for flow regression. ROI guidance improves model efficiency (as shown in **Table 3**) and suppresses the irrelevant background flow. We acknowledge that failed ROI detection can impact performance (**Sec. 5.6**) in challenging cases, such as blurry or some intersecting vessels in microvascular videos and low contrast or heavy occlusion in Spring videos. However, our model design mitigates the effect on flow estimation from different types of ROI errors:
> * **Over segmentation.** When an over-segmented ROI guides correlation lookup, some background pixels are also indexed and fed to subsequent steps. However, this doesn’t degrade final flow estimations, as the trained motion encoder and RNN further delineate the motion boundary through iterative flow updates. Full-frame flow regression could be considered as an extreme case of over-segmentation.
> * **Transient frames drop or incomplete ROI.** Motion memory module provides redundancy against occasional failures by maintaining estimation consistency with prior memories (**Sec. 3.3**). This feature stabilizes predictions when one frame has incomplete or no ROI detection. This is demonstrated by the ablation study of the motion memory module, where **FEPE dropped from 6.32 to 5.46 (Table 3)**. SAM2Flow incorporates an additional trick, known as warm start (**Supplement A**), to further stabilize the flow estimation in cases of occasional ROI instability.
> * **Missed/incomplete ROI over an extended period.** Failed ROI detection over more consecutive frames would result in failed optical flow estimation, since the model fails to identify any target. However, as an interactive pipeline, SAM2Flow offers the flexibility to add/correct user prompts for these challenging frames during inference. We recommend using multiple positive points spaced over the ROI for larger targets to mitigate prompt ambiguity.
>
> **Table 1 & 2**, along with **Figure 2**, show that SAM2Flow achieves the best foreground flow estimation accuracy and temporal stability, suggesting failure resilience even in challenging scenarios. ***We will clarify this failure-resilience design more explicitly in Sec. 5.6 (Limitations). And for the final version, we plan to include additional visual examples of flow estimation guided by the above three kinds of imperfect ROI detections in the supplement.***
> ## 3. How are foreground regions defined in the Spring dataset?
> Thank you for requesting more clarity. On the Spring dataset, foreground regions are defined as objects exhibiting significant relative motion to the background/camera movements. In each Spring video, the foreground ROIs are selected manually by reviewing the video and the paired optical flow ground truths.
>
> For each Spring video, we first manually select key frames for object detection. **Pre-trained SAM2-L** is applied for panoptic segmentation, prompted with a 16×16 grid of points across each selected frame. Among the generated masks, we manually remove tiny objects (e.g., small stones and water drops) and background (e.g., sky, trees). The retained masks are then propagated across the video by SAM2-L to generate a complete segmentation for each object in all frames. Finally, the generated ROIs are reviewed and manually refined to ensure segmentation accuracy and temporal consistency. In the training splits, we utilize all the segmented objects as ROIs to achieve maximum diversity of context and motion. For validation/test sets, we only test the models in 1-2 ROIs with the most prominent motions during evaluation. ***We will add these details of Spring dataset preparation in the revised Supplement B.3.***
> ## 4. How does SAM2Flow perform in complex scenarios, such as vascular overlapping or turbulent flow?
> **Structure Complexity.** Our manuscript specifically presents examples that include challenging crossing flow and vessel overlap (**Fig. 2**). These cases are common in clinical microvascular datasets. SAM2Flow is the only model that successfully separates the crossing flows, whereas SEA-RAFT and MemFlow incorrectly merge them. Our SAM2Flow model acheives this performance by utilizing:
> * **User prompts** guide the detection of separate ROIs (**Sec. 3.2**), preventing merging different flows into one.
> * **ROI-specific correlation lookup** allows separate flow regression for each vessel.
> * Propagation of **context and motion memories** maintains the identity of each vessel throughout the video.
>
> **Motion complexity.** Our dataset includes numerous cases of capillary loops that are abundant in the superficial oral mucosa. These vessels present drastic flow direction changes within short vessel segments (as shown in **Fig. 2**), thereby increasing flow complexity compared to long, flat laminar flows. While blood flow in most healthy blood vessels is approximately laminar, flow becomes turbulent at vessel branch points, vessel narrowing,  obstructions (such as in patients with sickle cell disease), or under certain conditions like hyperemia. An efficient flow estimation method like SAM2Flow could enable quantitative in vivo studies of microvascular flows under these clinically relevant conditions.
> ## 5. Is the method generalizable to other microvascular imaging techniques beyond OBM?
> We appreciate this critical discussion. OBM provides rich **phase contrast** at the single-cell level, which enables dense and continuous optical flow estimation. While our current dataset is based on OBM, the design of the SAM2Flow framework is **modality-agnostic**. In principle, it could be applied to other microvascular imaging techniques with high **spatial resolution** to identify blood flows and in-flow features, and sufficient **temporal resolution** to capture fast cell motions. For instance, high-speed fluorescent flow imaging [1] and other phase-contrast flow imaging, such as AOSLO [2], obtain very similar data. Several additional techniques, including Side-stream Dark Field (SDF) and Orthogonal Polarization Spectral (OPS), acquire blood flow data; however, these techniques rely on **absorption contrast** and produce temporally and spatially **sparse landmarks** for flow tracking (white blood cells and plasma gaps). As discussed in **Sec. 2.1**, it is particularly challenging to quantify these flows, particularly for conventional two-frame models, because the landmarks are not visible in every frame. The dual-memory module in SAM2Flow may mitigate such a limitation with enhanced spatial and temporal robustness.
>
> Generally, we believe SAM2Flow can generalize to other applicable modalities and various imaging sites, including the eyelid, nailfold, and retina, with available **annotated ground truths** and appropriate domain adaptation. We are actively exploring broader applications of SAM2Flow as future work. Moreover, we hope that by introducing the first optical flow neural network for blood flow analysis and sharing our dataset publicly, other groups will begin sharing their datasets and contributing to generalizable analysis models.
>
>     [1] Meng, Guanghan, et al. "Ultrafast two-photon fluorescence imaging of cerebral blood circulation in the mouse brain in vivo." Proceedings of the National Academy of Sciences 119.23 (2022): e2117346119.
>     [2] Joseph, Aby, et al. "Label-free imaging of immune cell dynamics in the living retina using adaptive optics." Elife 9 (2020): e60547.

---

### Official Review · Reviewer_No5r · 2025-07-01

**Clarity:** 3
**Significance:** 2
**Originality:** 3
**Rating:** 4
**Confidence:** 2

**Summary:**

The paper proposes a noel approach for optic flow estimation. Uniquely to the work, it focuses primarily on in vivo microcirculation analysis. Given a collection of images, the paper allows the user to select which area needs to have its optic flow computed via a prompting mechanism similar to the one in SAM. The rest of the work follows a relatively standard flow estimation pipeline, with the addition of a memory based mechanism for improving results on long video sequences.

**Questions:**

Overall I do not have significant suggestions for improvement, as I am very much a non expert in the specific application domain. From a pure CV optic flow estimation paper, the paper is not overwhelmingly novel, using concepts from SAM and relatively standard optic flow formulations. The application and dataset however appears to bring sufficient novelty to warrant publication.

If I had one request, it would be to anchor the design of the approach a bit more into the specific application. The main mention of multivascular flow estimation within the method appears to be on line 227, and this could be expanded.

**Ethical Concerns:**

["NO or VERY MINOR ethics concerns only"]

**Final Justification:**

Following the rebuttal and the other reviews, I maintain my overall positive view of the paper and believe it should be accepted.

**Limitations:**

Yes, these are discussed.

**Quality:**

3

**Strengths And Weaknesses:**

Positive:
- The paper is fairly well written and easy to understand.
- I am not an expert in the field, but according to the authors this is one of the first approaches to produce optic flow for this type application.
- The authors also introduce a novel dataset specific to microcirculation analysis.
- I very much appreciate that the authors included an application specific result section, commenting on the usefulness of the estimated flow.
- The paper also appears to produce good results for a more standard (i.e. non-medical) dataset.

Negative:
- The method section of the paper reads somewhat disconnected from the introduction and experimental section. The latter sections very much link to the target application domain, whereas the former seems quite generic i.e. not specific to microcirculation flow estimation.
- I am not an expert in the field so do not know if the methods evaluated are adequate for the specific task.

---

> ### Author Rebuttal · Authors · 2025-07-30
>
> We are grateful for the reviewer's appreciation of our novel application, the contribution of our new dataset, and the model’s generalizability. And we really appreciate your suggestions to enhance the motivations of model design. Below, we provide more details for your comments.
> ## 1. Anchoring the Method Section More Firmly in the Application Context
> During the design of the SAM2Flow framework, we tailor each component to address the clinical challenges of in vivo microvascular flow analysis. As suggested by the reviewer, ***we will revise the Method section*** to clarify our motivations further and explicitly link each module design to domain-specific challenges of microvascular flow estimation in the following ways:
> * **Prompt-conditioned Context (Sec. 3.2)**: prompt inputs enable user-defined vessel selection in cases where complex overlapping vessel structures or irrelevant background motion (e.g., heartbeat, tongue or jaw movement) can mislead global flow estimation. This is critical in OBM videos, where select vessels are more diagnostically relevant to clinicians.
> * **ROI-guided correlation lookup (Sec 3.2)**: As flows in OBM videos are highly localized, occupying only a small portion of the entire field of view, ROI-guided correlation lookup limits the search within the target vessels to increase computational efficiency and suppress the background noise in flow estimation. This enables practical application of SAM2Flow to match the high framerate (200 Hz) of OBM flow videos.
> * **Dual memory module (Sec. 3.3)**: Motivated by the need for temporal stability across long OBM videos (e.g., 12,000 frames for 60 seconds), SAM2Flow overcomes the limitation of conventional two-frame models by incorporating dual memory modules. The context memory helps to keep track of the identities of target vessels, while the motion memory ensures the long-term estimation smoothness for constantly pulsing flows, which is essential for downstream physiological analyses, such as flow waveform extraction as discussed in **Sec. 5.5**.
>
> Additionally, ***we will enhance the Introduction by clarifying the motivation of the model designs to tackle a real-world medical imaging challenge***.
>
> ## 2. SAM2Flow’s Novelty as a Generic Optical Flow Estimation Application
> We also note that SAM2Flow has demonstrated its novelty and superior performance for general applications (public datasets: **Spring in Table 2 & Sintel in Q3 of Rebuttal to Reviewer 33dS**). SAM2Flow offers unique features that are not present in previous optical flow models, including **flow stability over longer videos (Fig. 2)** and **User interaction** that enables Object/ROI-specific estimation. This would be a valuable contribution to various downstream tasks, such as motion detection and tracking, as well as object-specific video editing.

---

> > ### Comment · Reviewer_No5r · 2025-08-05
> >
> > Thank you for the rebuttal. I believe the paper should be accepted and maintain my positive review.

---

### Official Review · Reviewer_33dS · 2025-07-02

**Clarity:** 3
**Significance:** 3
**Originality:** 2
**Rating:** 4
**Confidence:** 5

**Summary:**

For microvascular blood flow, this work proposes an optical flow estimation method based on SAM2 to confirm the RoI (Region of Interest) area according to user prompts, and introduces a new microvascular blood flow dataset. The method achieves leading results on both standard optical flow benchmarks and this dataset.

**Questions:**

Please refer to the weaknesses.

**Ethical Concerns:**

["NO or VERY MINOR ethics concerns only"]

**Final Justification:**

The author has addressed most of my concerns in the rebuttal. Overall, I believe the significance of this work is worth accepting.

**Limitations:**

yes

**Paper Formatting Concerns:**

I did not notice significant formatting issues.

**Quality:**

3

**Strengths And Weaknesses:**

Strengths:

1. The method proposes an effective optical flow estimation approach for the microvascular blood flow task.

2. It provides a novel microvascular blood flow dataset.

3. The paper features good visualizations for demonstration.


Weaknesses:
1. Lack of comparison with more state-of-the-art optical flow estimation methods, such as the SAM-based method SAMFlow [1] and the multi-frame method StreamFlow [2].

2. Absence of results on more standard optical flow benchmarks (e.g., Sintel). Although the task focuses on microvascular blood flow, since there was no large-scale dataset for this task previously, experiments on more public benchmarks (whether fine-tuned results or 0-shot results) could more effectively validate the method’s effectiveness. After all, the method itself is not specifically designed for microvascular tasks, and the same architecture can be directly applied to optical flow estimation in other scenarios. More comparisons would better demonstrate its validity.

3. The experiments on the Spring dataset should present more metrics such as Fl.

[1] SAMFlow: Eliminating Any Fragmentation in Optical Flow with Segment Anything Model

[2] StreamFlow: Streamlined Multi-Frame Optical Flow Estimation for Video Sequences

---

> ### Author Rebuttal · Authors · 2025-07-31
>
> We thank the reviewer for the constructive review. We appreciate the recognition of the effectiveness of our method, the value of our novel dataset, and the strength of our visual demonstrations. We agree with the reviewer that the manuscript would be improved by the suggested additional experiments. In this rebuttal, we provide more details of some choices we made in the original study, and we are happy to share some new results of the suggested additional experiments.
> ## 1. Additional Comparative Models: SAMFlow and StreamFlow
> * **SAMFlow**: SAMFlow is a two-frame model that introduces the additional SAM encoder onto the FlowFormer backbone for enhanced estimation integrity per frame, with the challenge of partial occlusion. In our comparative study, FlowFormer++, a variant of FlowFormer, is the slowest model (**134 ms/frame vs/ 48.78 ms/frame of SAM2Flow**) in **Table 1**, while SAMFlow’s speed is similar to FlowFormer according to the original paper [1]. Although both SAMFlow and SAM2Flow leverage the foundation segmentation model SAM, our model design is tailored for efficient and temporally stable flow estimation. In contrast, the design motivation and excessive computational burden make SAMFlow impractical for our task on long, high-framerate microscopy videos (~200FPS, often >1 minute duration).
> * **StreamFlow**: We thank the reviewer for the suggestion to include StreamFlow in our comparative study. StreamFlow is an efficient multi-frame model that avoids recursive flow estimation with the streamlined in-batch multi-frame pipeline. Starting from the checkpoints pre-trained on “C+T+S+K+H” (same as the SEA-RAFT checkpoint, **supplement C.1**), we finetuned StreamFlow on our microvascular dataset, and evaluated it on the test split as follows:
> | Model | EPE_all | 1px | 3px | 5px | EPE_fg | 5px | 10px | 15px | Speed |
> |:---:|:---:|:---:|:---:|:---:|:---:|:---:|:---:|:---:|:---:|
> | SEA-RAFT | $\underline{1.28}$ (1.03) | **0.88** | $\underline{0.92}$ | $\underline{0.94}$ | $\underline{6.60}$ ($\underline{5.47}$) | **0.69** | **0.86** | $\underline{0.91}$ | **21.14** |
> | MemFlow$^{(MF)}$ | 1.79 (1.40) | **0.88** | 0.91 | 0.93 | 12.47 (10.23) | 0.58 | 0.74 | 0.80 | $\underline{43.98}$ |
> | StreamFlow$^{(MF)}$ | 1.43 ($\underline{1.02}$) | **0.88** | 0.90 | 0.93 | 10.13 (8.36) | 0.49 | 0.74 | 0.84 | 79.92 |
> | SAM2Flow$^{(MF)}$ | **1.14 (0.92)** | **0.88** | **0.93** | **0.96** | **5.84 (4.86)** | $\underline{0.66}$ | **0.86** | **0.93** | 48.78 |
> * $^{(MF)}$ denotes multi-frame models.
>
> The results indicate that StreamFlow outperforms MemFlow and achieves competitive accuracy in microvascular flow estimation. However, SAM2Flow remains superior in both accuracy (**FEPE=5.84 vs. 10.13**) and efficiency (**48.78 ms/frame vs. 79.92 ms/frame**). These results further highlight SAM2Flow’s unique ability to achieve high accuracy with impressive speed as a multi-frame model, which is critical for long, high-framerate microvascular videos. ***Additional results on StreamFlow on public benchmarks are presented in the following sections.***
>
>     [1] Zhou, Shili, et al. "Samflow: Eliminating any fragmentation in optical flow with segment anything model." Proceedings of the AAAI Conference on Artificial Intelligence. Vol. 38. No. 7. 2024.
> ## 2.  Additional Metrics for Spring Benchmark
> To provide a more comprehensive view of each model’s performance, we follow the previous works on the Spring dataset and add the following metric, with the standard definition and thresholds from the original Spring dataset, to expand the current **Table 2**:
> * **Fl-fg**: flow outlier (>3pxs and > 5% of GT flow magnitude) rate for the foreground regions;
> * **Fl-epe**: the average EPE of the flow outliers from the foreground regions;
> * **WAUC-fg**: weighted AUC for estimations on foreground pixels.
> | Model | FEPE | 1px (%) | Fl-fg (%) | Fl-epe | WAUC-fg |
> |:---:|:---:|:---:|:---:|:---:|:---:|
> | RAFT | 2.25 (6.97) | 74.25 | 9.87 | 7.98 (8.98) | 74.76 |
> | GMA | 2.17 (6.30) | 76.21 | 9.08 | $\underline{7.67}$ ($\underline{7.72}$) | 79.67 |
> | SEA-RAFT | $\underline{1.45}$ ($\underline{5.61}$) | 86.32 | **5.18** | 8.18 (10.85) | $\underline{83.85}$ |
> | MemFlow$^{(MF)}$ | 1.56(7.27) | $\underline{86.49}$ | 7.24 | 8.82 (12.72) | 83.54 |
> | StreamFlow$^{(MF)}$ | 1.54 (7.23) | 85.16 | 5.82 | 7.74 (10.16) | 82.51 |
> | SAM2Flow$^{(MF)}$ | **1.23** (**4.21**) | **87.13** | $\underline{5.29}$ | **7.30** (**5.29**) | **84.57** |
>
> These additional metrics provide better insights into model performance regarding estimation outliers. For the estimation regarding the ROIs, SAM2Flow achieves the best fine-grained accuracy (highest 1px, **87.13%**) and most robust estimation, with fewer flow outliers (**Fl-fg=5.29%**), and lowest **Fl-epe (7.30)**. SEA-RAFT performs slightly better in **Fl-fg (5.18%)** but underperforms on **Fl-epe** and **WAUC-fg**, suggesting less stable estimation across the ROIs with some extreme outliers.
>
> ## 3. Additional Benchmark: Sintel
> To study the generalizability of SAM2Flow on non-medical scenarios, we have reported results on the Spring dataset. Both the Sintel and Spring datasets consist of animation sequences. We prioritized Spring because it contains more (**37 vs. 23**) and longer (**270 vs. 45 frames/video**) videos compared to Sintel (as shown in the **Supplement table 4**). Intuitively, such long sequences enable better demonstration of the robust optical flow estimation empowered by the proposed dual-memory module of SAM2Flow. We agree that demonstrating the generality of SAM2Flow on additional benchmarks like Sintel (Clean & Final versions) would strengthen the impact. Similar to Spring, we prepare the dataset with additional annotations of foregrounds. The dataset details will be added to the **Supplements** as **B.4**. We split the total of 23 scenes into training (14), validation (3), and test (6). The training and validation combine both **Clean** and **Final** videos. For evaluation, we report the performance on the Clean and Final videos separately.
>
> For fair comparison, we start the experiments with the checkpoint pre-trained on the two-frame **Flying-Chair** and **Fly-Things** datasets for all comparative models and SAM2Flow backbones. Next, we train all the models with the prepared **Spring dataset**, which is especially helpful for multi-frame models to incorporate long-term temporal information into learning. Lastly, all the models are finetuned on the Sintel training split. For evaluation, we report the **foreground end-point error (FEPE)** on both clean and final test splits following previous works. The preliminary results are shown in the following table:
>
> | Model | FEPE-Clean | FEPE-Final |
> |:---:|:---:|:---:|
> | RAFT | 5.21 (9.30)  | 5.47 (10.12)  |
> | SEA-RAFT | $\underline{3.26}$ (7.54) | $\underline{4.08}$ (8.94) |
> | StreamFlow$^{(MF)}$ | 4.06 (5.37) | 4.43 (5.66) |
> | SAM2Flow$^{(MF)}$ | **3.17** (6.89) | **3.39** (5.97) |
>
> As for now, we have finished the training and evaluation of the best-performing models, based on the results from the Microvascular and Spring datasets. ***We are currently training more models, including GMA and MemFlow, to complete this table. The complete results will be updated in the final draft.*** For the current results, SAM2Flow consistently demonstrates superior generalization and robustness compared to all baselines on Sintel, particularly on the challenging Final video, where more textures are added to the scene. **These results, combined with the strong performance on the Spring dataset, highlight that SAM2Flow achieves the state-of-the-art ROI-centric optical flow estimation beyond the microscopic domain.**

---

> ### Author Response · Authors · 2025-08-04
>
> Dear Reviewer 33dS,
>
> Thank you for reading our rebuttal. As promised in the rebuttal, we have completed the comparative study on the **Sintel** dataset, with two additional comparative models (GMA and MemFlow). Please see the full results in the table below.
>
> | Model | FEPE-Clean | FEPE-Final |
> |:---:|:---:|:---:|
> | RAFT | 5.21 (9.30)  | 5.47 (10.12)  |
> | GMA | 4.65 (7.45) | 5.14 (8.17) |
> | SEA-RAFT | $\underline{3.26}$ (7.54) | $\underline{4.08}$ (8.94) |
> | MemFlow$^{(MF)}$ | 3.77 (5.82) | 4.27 (7.09) |
> | StreamFlow$^{(MF)}$ | 4.06 (5.37) | 4.43 (5.66) |
> | SAM2Flow$^{(MF)}$ | **3.17** (6.89) | **3.39** (5.97) |
>
> We sincerely appreciate your insightful inputs that help improve our manuscript. **Please let us know if there is any additional information we could provide during the rest of the discussion period to facilitate your final decision.**
>
> Thanks,
>
> Authors of Submission 3465

---

> > ### Comment · Reviewer_33dS · 2025-08-06
> >
> > Thank you to the authors for the additional validation. These validations have well addressed my concerns. However, I believe the authors should double-check the reported speed of StreamFlow in the supplementary experiments. Did you use the non-overlapping inference pipe?

---

> ### Author Response · Authors · 2025-08-06
> **Correction of StreamFlow speed**
>
> Dear Reviewer 33dS,
>
> Thank you so much for pointing this out. We just double-checked the StreamFlow inference. We used the non-overlapping inference pipeline. The numbers of iterations and frames were set to be 12 and 4, following the original paper. And the average inference speed should be corrected to 60.07 ms/frame.
>
> We were running multiple additional training sessions at the same time during the rebuttal session. And there was another model training on the same GPU while we ran the test on StreamFlow, which might have slowed down the StreamFlow inference speed. For all other inference speed tests, we ran the models on a single GPU that was totally free. And we just reran the StreamFlow test on a free GPU for fair comparison. Please pardon our honest mistake, and we will update the correct results in the final manuscript. Thank you again for the correction.
>
> Thanks,
>
> Authors of Submission 3465

---

### Author Response · Authors · 2025-08-08
**Thank you for your reviews and feedback**

Dear Reviewers,

We are sincerely grateful for your insightful reviews and prompt feedback during the discussion session. Your inputs significantly improved this manuscript and inspired our future work. We hope our rebuttals and responses adequately address your concerns and are helpful for your final decisions. **As this discussion period is coming to a close, we are here for any additional comments or questions.**

We wish you the best in your future research and thank you for such a meaningful experience at NeurIPS.

Thanks,

Authors of submission 3465

---

### Note · Authors · 2025-08-11

Dear AC,

We sincerely appreciate your hard work supporting the review process. We received four high-quality reviews and prepared a detailed rebuttal for each one. According to the responses from all four reviewers, our rebuttals have well addressed all the discussions and concerns raised in the reviews. Here, we provide brief summaries of proposed work and the discussions to facilitate your final decision.

## Summary of the proposed SAM2Flow model

In this work, we propose SAM2Flow, a novel interactive optical flow estimation model for long microvascular flow videos. The major contributions are as follows:
* **Interactive foreground optical flow estimation.** Our model enables the first ROI-centric optical flow estimation, achieving ROI tracking and refined foreground flow estimation with sparse user prompt inputs.
* **Dual memories for long-term estimation.** SAM2Flow achieves the most stable flow estimation over long videos, with the context and motion memory module.
* **Strong generalizability.** Experiments on two public datasets suggested SAM2Flow’s potential for generic optical flow estimation.
* **Novel flow dataset.** We also proposed an in vivo microvascular dataset with a large number of long videos.

We thank all four reviewers for acknowledging the clinical importance of in-vivo blood flow estimation, the novelty and generalizability of SAM2Flow, and the contribution of our microvascular dataset.
## Summary of discussion
* Thanks to Reviewer 33dS, we have included **an additional multi-frame comparative model (StreamFlow)**, conducted experiments on **an additional public benchmark (Sintel)**, and added **additional metrics** for the Spring benchmark. These additional experiments showed consistent results and further strengthen our conclusion that SAM2Flow demonstrates superior foreground optical flow estimation accuracy in all datasets, with competitive efficiency.
* Thanks to Reviewer 5zif and D1JP, we expanded our discussion on **the reliability of ROI** and elaborated on **how the flow estimation would be affected by different types of ROI detection errors**. This discussion provides a better view of the model's performance in various challenging scenes. We have promised to add visual examples to the supplement for the final version.
* Thanks to Reviewer No5r, we have clarified **the motivations behind the design of SAM2Flow modules** and the corresponding relevance to long-term microvascular flow.

Thanks,

Authors of submission 3465

---

### Decision · Program_Chairs · 2025-09-17

**Decision:**

Accept (poster)

**Comment:**

All reviewers vote for accepting the paper. They state that it provides an "effective optical flow estimation approach for the microvascular blood flow task" with a "novel microvascular blood flow dataset" (33dS), the "paper is fairly well written and easy to understand" (No5r), "The studied problem is important", and "sharing the dataset is a good contribution" (D1JP). Therefore the AC reccomends acceptance of the paper.